# Genomic Analysis of Amphioxus Reveals a Wide Range of Fragments Homologous to Viral Sequences

**DOI:** 10.3390/v15040909

**Published:** 2023-03-31

**Authors:** Qiao Du, Fang Peng, Qing Xiong, Kejin Xu, Kevin Yi Yang, Mingqiang Wang, Zhitian Wu, Shanying Li, Xiaorui Cheng, Xinjie Rao, Yuyouye Wang, Stephen Kwok-Wing Tsui, Xi Zeng

**Affiliations:** 1Agricultural Bioinformatics Key Laboratory of Hubei Province and 3D Genomics Research Centre, College of Informatics, Huazhong Agricultural University, Wuhan 430070, China; qiaodu_scu@126.com (Q.D.); fpeng_study@163.com (F.P.); kejin.xu@phasesci.com (K.X.); zhitian.wu@wur.nl (Z.W.); lishanying2022@163.com (S.L.); 17265654963@163.com (X.C.); rebecca8130@163.com (X.R.); yx_wang128@163.com (Y.W.); 2School of Biomedical Sciences, The Chinese University of Hong Kong, Hong Kong, China; qingxiong@cuhk.edu.hk (Q.X.); styangyi@live.cn (K.Y.Y.); mqwang@link.cuhk.edu.hk (M.W.); kwtsui@cuhk.edu.hk (S.K.-W.T.); 3Hong Kong Bioinformatics Centre, The Chinese University of Hong Kong, Hong Kong, China; 4Stanford Cardiovascular Institute, Stanford University School of Medicine, Stanford, CA 94305, USA

**Keywords:** amphioxus, virus, homologous fragment, evolution

## Abstract

Amphioxus species are considered living fossils and are important in the evolutionary study of chordates and vertebrates. To explore viral homologous sequences, a high-quality annotated genome of the Beihai amphioxus (*Branchiostoma belcheri* beihai) was examined using virus sequence queries. In this study, 347 homologous fragments (HFs) of viruses were identified in the genome of *B. belcheri* beihai, of which most were observed on 21 genome assembly scaffolds. HFs were preferentially located within protein-coding genes, particularly in their CDS regions and promoters. A range of amphioxus genes with a high frequency of HFs is proposed, including histone-related genes that are homologous to the *Histone* or *Histone H2B* domains of viruses. Together, this comprehensive analysis of viral HFs provides insights into the neglected role of viral integration in the evolution of amphioxus.

## 1. Introduction

Many studies involving viral homologous sequences in the genomes of higher animals have been reported, and sequences homologous to viral oncogenes have been identified in most vertebrates [1,2,3]. For example, it has been demonstrated that a sequence in the human genome is homologous to the v-*myc* oncogene of the *Avian myelocytomatosis virus* [4], and wild bats host sequences homologous to sequences in various eukaryotic viruses [5]. The analysis of the viral homologous sequences in animals helps in revealing the occurrence of endogenous viral integration and viral invasion, as well as genetic material exchanges between viruses and hosts [5,6,7,8,9]. The endogenous integration of the virus into higher animal genomes is believed to play an important role in evolution [10,11]. Endogenous retroviruses (ERVs) shape the evolution of transcriptional networks, and some ERVs can encode complete proteins and play physiological functions within hosts [12,13]. Endogenous bornavirus-like elements and some other nonretroviral endogenous viral elements encode functional proteins in host animals [10,14,15]. It is noteworthy that the diversity and evolution of the viruses themselves are also affected by endogenous integration [16]. Thus, viral homologous sequences provide valuable information regarding evolutionary modifications associated with hosts as well as virus–host interactions.

The common histones H2A, H2B, H3, and H4 are quintessentially eukaryotic proteins. Nevertheless, the homologs of the genes that encode these eukaryotic histones have also been identified in the DNA of virus genomes. *Pandoravirus* genomes have histone genes with eukaryotic homologs [5], and a *Pandoravirus*-related genome assembled from the marine metagenome encodes an H4-like protein that is 77% identical to human H4 [17]. Viruses in *Marseilleviridae* and the medusavirus in the order Pandoravirales encode distinct histone-like proteins homologous to all four core eukaryotic histones [18,19,20]. Some viral histone genes alter host gene expression by expression in the host, leading to negative changes in host development and metabolism that are beneficial to the viruses [21,22,23,24]. Viral histone H4 alters host gene expression by interacting with eukaryotic nucleosomes [23]. Bracoviruses have been shown to use their histone genes as weapons to gain an advantage by suppressing host immune responses and development [25,26,27]. Many viral histones have not been investigated to identify their functions and evolutionary roles in hosts and viruses. The group of basal chordates, amphioxus, contains some of the closest living invertebrate relatives of vertebrates [28,29,30,31,32,33]. Its members play a pivotal role in elucidating the evolution of chordates and vertebrates [34,35,36]. However, there have been few systematical studies of the viral homologous sequences and histones in amphioxus species.

A genomic analysis was performed to fully identify the viral homologous sequences in amphioxi. In this study, the adult Beihai amphioxus was suggested as a *B. belcheri* subspecies and was tentatively named *B. belcheri* beihai. A high-quality *B. belcheri* beihai genome was assembled and annotated by our team [37]. A total of 347 HFs in the *B. belcheri* beihai genome were identified, and the genomic features of the HFs were analyzed. A few amphioxus genes with high-confidence HFs and genomic regions enriched with HFs were proposed. More importantly, there were highly conserved sequences in histone-related genes between viruses and the amphioxus. The analysis of the viral HFs in the amphioxus genome could provide abundant evidence for endogenous integrations, broadening our understanding of the evolution of and interaction between amphioxus species and viruses.

## 2. Materials and Methods

### 2.1. Sample Collection

Adults of the Beihai amphioxus, *B. belcheri* beihai were obtained from the sea near Dianbai District, Maoming City, Guangdong Province, China, cultured at 24–28 °C with air-pumped circulating artificial seawater in Beihai Marine Station of Nanjing University in Beihai City, Guangxi Province, China, and fed with seawater and sea alga.

### 2.2. Genome Sequencing and De Novo Genome Assembly

Genome sequencing and de novo genome assembly of *B. belcheri* beihai has been reported [37]. In brief, the genomic DNA was sequenced by Illumina HiSeq 2000 in 500-bp and 3000-bp libraries to generate paired-end NGS short reads and PacBio RS system for third-generation long reads. The first draft of genome assembly was constructed with PacBio long reads using Canu v1.8 [38]. Then, the final genome assembly was generated after scaffolding and polishing of the draft assembly. The continuity of the genome assembly was assessed by QUAST v5.0.2 [39].The completeness was assessed by BUSCO v3.1.0 [40] with database metazoa_odb9.

The genome assembly size of *B. belcheri* beihai was 478,319,013 bp and the scaffold number was 583. The scaffold N50 length was 4,185,906 bp and the gap content was 0.156%. Regardless of the gaps, the ungapped length of *B. belcheri* beihai was 474,945,525 bp. As for completeness, the genome assembly of *B. belcheri* beihai was 97.2%.

### 2.3. Genome Annotation

Firstly, repeat masking was performed by de novo prediction with RepeatModeler v2.0.1 [41], and masking with RepeatMasker v4.0.8 (RepBase edition 20181026) [42]. In the de novo prediction with RepeatModeler v2.0.1 [41], the prediction of repeat families in the genome was performed using RECON v1.05 [43] and RepeatScout v1.0.6 [44]. Then, Maker pipeline v2.31 was used for genome annotation [45]. In the Maker pipeline, the alignment was supported by transcriptome assemblies and homologous proteins by Exonerate v2.4.0 [46], whereas gene prediction was accomplished by SNAP (lib v2017–03-01) [47], GeneMark v4.38 [48] and Augustus v3.3.1 [49]. The quality of genome annotation was assessed by BUSCO v3.1.0 [40] with database metazoa_odb9. The completeness of genome annotation was 95.2%. In total, 44,745 protein-coding genes were annotated in *B. belcheri* beihai, and most of genes were homologous to *Homo sapiens* and *Mus musculus* genes (Appendix A). 

### 2.4. Identification of HFs in Amphioxus Genome

A total of 9569 viral genomes were searched in the genome of *B. belcheri* beihai by BLASTN v2.5.0 [50] at E-value cutoff of 1 × 10^−5^ to obtain 363 aligned fragments. The *B. belcheri* beihai genome was used as the reference sequence to build the library. The aligned fragments were filtered to obtain HFs. 

A fragment of viral sequences might align with multiple DNA fragments in the *B. belcheri* beihai genome. There were also fragments of viral sequences that only aligned with a fragment of *B. belcheri* beihai. To make the essence of the alignment score the same, the HFs of *B. belcheri* beihai that were in line with the first situation were retained. There was another situation of alignments where multiple fragments of viral sequences aligned with the same DNA fragment in the *B. belcheri* beihai genome. The results with the minimum E-value were retained after filtering using the E-value cutoff, and finally 347 HFs were identified in the *B. belcheri* beihai genome.

### 2.5. Identification of HFs in Viral Genomes

Multiple fragments of viral sequences were less likely to align with the same DNA fragment in the *B. belcheri* beihai genome. BLASTN results with the minimum E-value after filtering using the E-value cutoff of 1 × 10^−5^ were retained, and all results with the same minimum E-value were retained. In total, 361 results were obtained from 363 alignment results through above processing. The same fragment of viral sequences aligned with multiple DNA fragments in the *B. belcheri* beihai genome. In this case, 69 results with the minimum E-value were retained. Because there were adjacent breakpoints in the viral genomes, a threshold of 10 bp was adopted to merge the adjacent fragments into a single fragment. The duplicate DNA fragments were removed after merging, and a total of 50 HFs were identified in 17 viral genomes.

### 2.6. Data Analysis

The data analysis statistics were mainly implemented by Linux shell v4.2.46 and R v4.0.5 [51] programming, and the ggplot2 [52] v3.3.6 R package undertook most of visualization tasks in this study. Statistical significance was assessed by computing either the *p* value or the adjusted *p* value (using the Bonferroni method) based on the chi-squared ($X^{2}$) test. This test was selected because it enables comparison of the observed frequency distribution of a categorical variable with an expected frequency distribution that follows a specific theoretical distribution, as previously reported in the literature [53,54,55,56].

The simplified pipeline, the schematic diagram for the HFs and the schematic indicating that multiple fragments of viral DNA were homologous to one fragment in the *B. belcheri* beihai genome were manually drawn in PowerPoint 2019. The schematic diagram for HFs containing the longest length with the highest alignment quality were also manually drawn through Illustrator for Biological Sequences v1.0 [57,58] online. The Circos image for the gene analysis and functional annotation of the viral HFs was completed by using the OmicStudio tools at https://www.omicstudio.cn/tool (accessed on 22 November 2022). The images for general profiles of HFs and distribution of HFs in 100 kb sliding windows were generated in SVG format based on Perl v5.10 (http://www.perl.org/, accessed on 25 September 2022). 

## 3. Results

### 3.1. Identification of Viral HFs in the Amphioxus Genome 

To search for viral HFs, viral sequences were mapped to the *B. belcheri* beihai genome and 347 HFs between *B. belcheri* beihai and 17 viral genomes were obtained after filtering (Table 1, Figure 1 and Figure 2). The average length of these HFs was 174 bp (range 33–277 bp). The viruses were from *Pandoravirus*, *Bracovirus*, *Bat associated circovirus 4*, *Choristoneura fumiferana granulovirus*, *Betaretrovirus*, *Alpharetrovirus*, *herpesvirus*, *Pygoscelis adeliae polyomavirus 1*, *Myoviridae* and *Phycodnaviridae*. *Y73 sarcoma virus* and *Mason-Pfizer monkey virus* (*MPMV*) are retroviruses. The viruses with the most HFs with the amphioxus belonged to the genus *Pandoravirus* with large genomes and morphologies. A total of 172 HFs from three pandoraviruses were observed. Notably, aquatic viruses and herpesvirus, in this study, were homologous with *B. belcheri* beihai, which is consistent with previously published findings. 

To decode the HF features in the *B. belcheri* beihai genome, we surveyed the distribution of HFs. The HFs were not uniformly distributed on 57 of the 583 *B. belcheri* beihai genome scaffolds. The HFs were observed to be enriched on 21 scaffolds (adjusted *p* value < 0.05, $X^{2}$ test) (Figure 3a), suggesting that the location preference of HFs exists at the scaffold level. The average length of these scaffolds was 1,696,183 bp (range 29,688–6,451,831 bp). On each scaffold enriched with HFs, an average of five genes was observed, each of which contained nine HFs on average. The most genes (22) with HFs and the most HFs (43) were identified on scaffold 62. There were no genes with HFs on scaffolds 100, 221, 271, and 336. Notably, the majority of the genes on the scaffolds enriched with HFs were related to histone. 

The analysis of the distance of neighboring HFs across the *B. belcheri* beihai genome further demonstrates the prevalence of HFs. The HFs were significantly enriched near each other (*p* < 0.01, $X^{2}$ test) (Figure 3b). Strikingly, 43.0% of HFs were located within 1 kilobase (kb) of one another (*p* = 1.471366 × 10^−41^). Subsequently, the enrichment of HFs in 100 kb sliding windows in the *B. belcheri* beihai genome was verified, and it was discovered that the genomic windows enriched with HFs were primarily distributed in scaffolds 1, 14, 25, 30, 41, 48, 62, and 64 (Figure 3c). A total of 59 amphioxus genes with HFs were observed in the genomic regions enriched with HFs. Of these, 91.5% were primarily associated with histone. We propose that a potential preference for HFs exists at selective target genes.

Among the HFs in the *B. belcheri* beihai genome, 56.5% were in the CDS regions, 36.3% were in introns, and 46.7% were in the promoter region. Statistically significant enrichment of HFs in the genes, CDS, and promoter regions of *B. belcheri* beihai was noted (Figure 3d; *p* < 0.05, $X^{2}$ test). Notably, the enrichment of HFs in promoters underscores the potential influence of HFs on the transcription of specific genes.

Next, the HFs in viral genomes were investigated. A total of 50 HFs in 17 viral genomes were identified (Appendix A), suggesting the influence of the repeat content in the *B. belcheri* beihai genome on the homologous count. Another 21 HFs were observed in the CDS regions of the viral genomes (Appendix A). However, the observed numbers of HFs in the CDS regions were significantly less than the expected numbers (*p* < 0.01, $X^{2}$ test) (Appendix A), suggesting that HFs are not prone to be located in CDS regions in viral genomes. The five viruses with the greatest number of HFs in the viral genomes were *Cyprinid herpesvirus 1, Pandoravirus* spp. *(Pandoravirus dulcis and Pandoravirus inopinatum), Choristoneura occidentalis granulovirus*, *Cotesia congregata bracovirus* and *Equid gammaherpesvirus 5*. The five viruses with the greatest number of HFs in the amphioxus genomes were *Pandoravirus* spp. (*P. dulcis* and *P. inopinatum*), *C. congregata virus*, *Tadarida brasiliensis circovirus 1*, *C. occidentalis granulovirus* and *MPMV*, among which pandoraviruses had the largest genome in the various environments [59]. *Tadarida brasiliensis circovirus 1*, which was detected in a bat species taxonomically, is a new species in the genus *Circovirus* [60]. *Circoviridae* viruses have been reported to infect many vertebrates [61]. *MPMV* is a primate retrovirus and can encode a protease. Viral structure proteins and viral enzymes are formed by processing virus-encoded polyprotein precursors through this protease [62,63].

### 3.2. Gene and Functional Annotation of the Viral HFs

Within the annotated genomes of *B. belcheri* beihai, 36 genes with 286 HFs were identified and analyzed. A number of hot-spot genes with HFs were discovered (Figure 4a). The 10 genes with the most HFs were *Histone H2B 1/2*, *Histone H4*, *Late histone H2B.2.1*, *Transposon TX1 uncharacterized 149 kDa protein*, *Histone H2B (Fragments)*, *DCST1*, *H2BC13*, *DCST2*, *hist2h2l and Histone H2B* (Appendix A), of which seven genes were histone related. There were 14 HFs in the Transposon TX1-related gene, encoding the uncharacterized 149 kDa protein, Transposon TX1. Interspersed repeats were transposable elements divided into DNA transposons and retrotransposons. Notably, the repeat contents of *B. belcheri* beihai were 37.21%, of which most repeats were interspersed repeats.

In the analysis of the general profile of HFs at the gene level, *P. dulcis*, *P. inopinatum*, *C. congregata bracovirus* and *Pandoravirus salinus* were homologous to four histone-related genes in five hot-spot genes, with most HFs being found in the amphioxus genome (Figure 4b). *T. brasiliensis circovirus 1* was homologous to the Transposon TX1-related gene. The DNA fragments homologous to *P. dulcis*, *P. inopinatum*, *C. congregata bracovirus* and *P. salinus* were in the promoters, CDS, and introns of the histone-related genes of the amphioxus, further denoting the role of these HFs in specific gene expression and transcription. Although the majority of the HFs in the *B. belcheri* beihai genome were homologous to only one DNA fragment in the viral genome (Appendix A), some HFs or some genes could be homologous to multiple viral DNA fragments (Figure 4a,b). Of the HFs in the *B. belcheri* beihai genome, 6.90% and 2.60% were homologous to either two or three viral genome DNA fragments, respectively. The number of HFs within the genes and corresponding viral species were mostly distinct (Figure 4a), indicating the diversity and heterogeneity of HFs. Therefore, we propose that there is an association between viruses and the amphioxus at the individual gene level. 

Subsequently, 18 amphioxus genes with HFs in CDS regions were identified, of which 13 genes were related to histones (Appendix A). A total of 193 HFs in 13 histone-associated genes were noted, and the average length of these HFs was 245 bp (range 153–276 bp). These HFs accounted for an average proportion of 50.1% of the length of these histone-related genes (range 0.500–88.4%). A total of 193 HFs were from *C. congregata bracovirus*, *P. dulcis*, *P. salinus* and *P. inopinatum* (Table 2). Among these four, except for *P. inopinatum*, the other viruses contained DNA fragments homologous to amphioxus HFs within their *Histone or Histone H2B* domain (Table 2). A total of 59.0% HFs (23/39) between the histone-related genes of the amphioxus and *C. congregata bracovirus* accounted for >50.0% of both of their lengths. In the amphioxus, 64.3% HFs (45/70) accounted for >50.0% of histone-related gene length but accounted for an average of only 25.7% of the *Histone H2B* domain-containing protein gene length *in P. dulcis*. 

Sequence alignment analysis allowed us to focus on the HFs with high confidence of high alignment quality and long length. Notably, the five HFs with the highest confidence (271–276 bp) were all located in *Histone H4* and *Histone H2B* on scaffold 14 (Table 3). The high-confidence HFs within *Histone H4* were located in the exon region (Figure 4c). When inspecting the virus side, these five amphioxus HFs were all homologous to the CDS regions of the *Histone* of *C. congregata bracovirus*. Additionally, *P. inopinatum* and *P. dulcis* contained DNA fragments homologous to the same *Histone H4* and *Histone H2B* on scaffold 14 of the amphioxus, although these HFs were not among the five HFs (Figure 4c and Appendix A). Specifically, two amphioxus HFs within the promoter of *Histone H4* were homologous to the intergenic and CDS regions of a hypothetical *P. inopinatum* gene and the CDS region of the *Histone H2B* domain-containing protein of *P. dulcis*, respectively (Figure 4c and Appendix A). Notably, the viral DNA fragments within the CDS region of the *Histone H2B* domain-containing protein in *P. dulcis* were homologous with the DNA fragments within the exon region of *Histone H2B* in the amphioxus (Figure 4c and Appendix A). In summary, highly conserved sequences of histone-related genes between the amphioxus and viruses were observed.

## 4. Discussion

As a crucial group of invertebrate chordates, amphioxus is considered an appropriate subject for studying the evolution of vertebrates and chordates. HFs are viewed as partial evidence of viral endogenous integration, which plays a key role in host genome evolution. However, the HFs of viruses in the amphioxus genome are not yet fully understood. Our study identified 347 confident HFs of viral sequences in the genome of Beihai amphioxus, *B. belcheri* beihai. We comprehensively investigated viral HFs in the *B. belcheri* beihai genome using the annotated genomes of amphioxus and viruses. The investigation revealed the preference distribution of HFs and the list of related genes. 

We identified 17 viruses that are homologous to amphioxus, including aquatic viruses and herpesvirus. These findings are consistent with previous publications that suggest that aquatic viruses and herpesvirus have integrated into the amphioxus genome [6,64,65,66]. The habitat of amphioxus is typically in temperate or tropical ocean [67], which could facilitate the fusion of aquatic viruses with the amphioxus genome. Among the 10 genes in amphioxus with the most HFs, the Transposon TX1-related genes, *DCST1* and *DCST2* were identified, in addition to seven histone-related genes. Transposable elements (TEs) have been discovered in the amphioxus genome [68], and some TEs in animals have been reported to have homology with viruses [69,70]. *DCST1* has been identified as a regulator of Type I interferon signaling through its interaction with STAT2 [71]. Type I interferon mediates the innate immune response to control virus infections in invertebrates [71,72]. New members of the signal transducer and activator of transcription (STAT) family have been reported in the chordate amphioxus, which can exert similar biological functions to vertebrate STATs [73]. *DCST2* is an important paralog of *DCST1*. A significant proportion of HFs showed a clear preference for genes, specifically CDS and promoter regions. Furthermore, the histone-related genes with a high frequency of HFs were homologous to the *Histone* or *Histone H2B* domain of viruses.

We suggest that most of these HFs possibly resulted from ancient integrations of viral DNA into the amphioxus genome. The functional and evolutionary influence of endogenous viral integration into the host genome has been reported. Consistent with previous conclusions, and our results, we speculate that viruses could interfere with the expression of genes in *B. belcheri* beihai, especially *Histone H4* and *Histone H2B*, through changes in transcript structure or cis-regulatory patterns [74,75,76,77]. Along with other published findings, and our results [78,79], viruses appear to hijack genes or manipulate the mechanism of *B. belcheri* beihai to help themselves achieve gene expression, favoring their survival and establishing dominance in the host. This may be because viruses required various proteins to favor their own survival in ancient times when the fusions or integrations occurred. In this study, the sequences of histone-related genes of *P. dulcis*, *C. congregata bracovirus* and *P. salinus* are highly homologous to those of amphioxus, which could play an important role in the origin of the nucleus of modern eukaryotic cells [80]. Viral HFs were also searched for in genomes of *Erpetoichthys calabaricus* (Vertebrata)**,**
*Aplidium turbinatum* (Urochordata), *Lytechinus variegatus* (Echinodermata), *Saccoglossus kowalevskii* (Hemichordate), *Drosophila melanogaster* (Protostomia), *Dendronephthya gigantea* (Cnidaria) and *Amphimedon queenslandica* (Porifera) using discontiguous megablast of NCBI’s BLASTN programs. Many of the viral HFs were also present in the genomes of these animal species, but the matching similarities were mostly lower than those observed in amphioxus regarding the bit-scores and identities. Therefore, we speculate that part of the viral HFs may not have entered the amphioxus genome directly through endogenous integration events, but had passed through several species during evolution, finally being inherited into the amphioxus genome. A large number of viral histones and their similarity to eukaryotic histones have not been thoroughly investigated, and further research is needed to better understand the origins of viral histones and their relationship with eukaryotes. Although the transcriptome data and amino acid sequences of amphioxus were not analyzed in this study, and we did not perform experimental validation of candidate integrations or confirm the timing of the integration events, we expect that this research will make a significant contribution to the future study of eukaryotic and viral evolution.

We believe that further investigation of viral homologous sequences in amphioxus and other animal genomes, particularly histone genes, will be valuable for advancing our understanding of the evolution of amphioxus and viruses. Together, our study offers an overview of the homologous locations and annotations shared between the genomes of amphioxus and viruses. We propose that the viral genomic elements identified in the amphioxus genome in this study may have played a crucial role in the evolution of both amphioxus and viruses, particularly in relation to histone genes. This study sheds light on the ancient history and evolution of these organisms and provides a foundation for further research in this area.

## Figures and Tables

**Figure 1 viruses-15-00909-f001:**
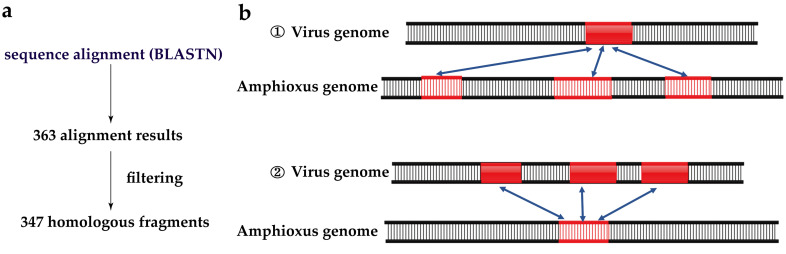
The detection method for HFs and the schematic diagram for HFs. (**a**) The simplified pipeline for detecting HFs. (**b**) The concrete schematic diagram for HFs. There were 2 situations of alignment. A fragment of viral sequences aligned with multiple DNA fragments in the *B. belcheri* beihai genome. Multiple fragments of viral sequences aligned with the same DNA fragment in the *B. belcheri* beihai genome.

**Figure 2 viruses-15-00909-f002:**
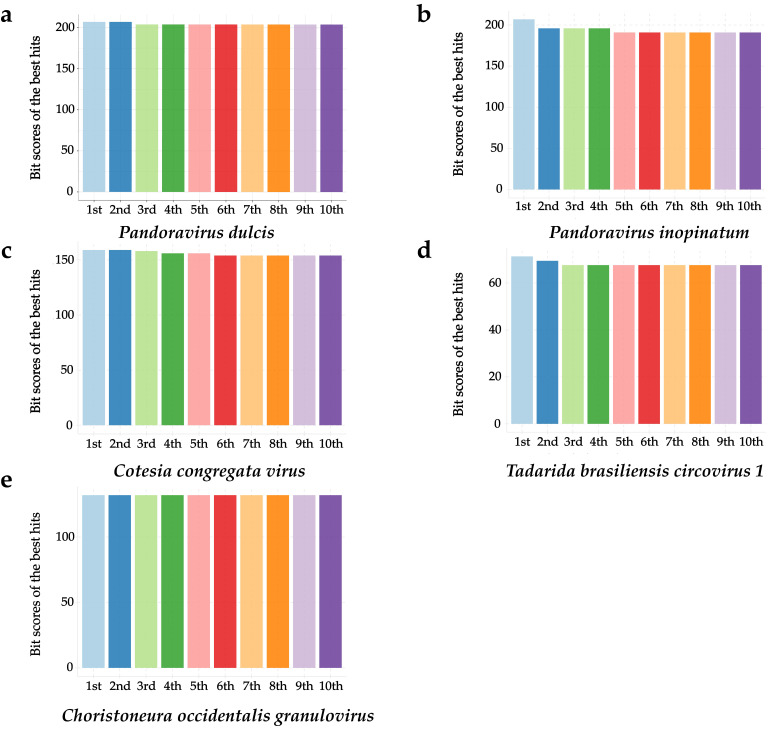
The alignment quality of 10 HFs in 5 virus species with the most HFs in the *B. belcheri* beihai genome. The averages of the Bit scores of the 10 HFs with the smallest E-values were shown.

**Figure 3 viruses-15-00909-f003:**
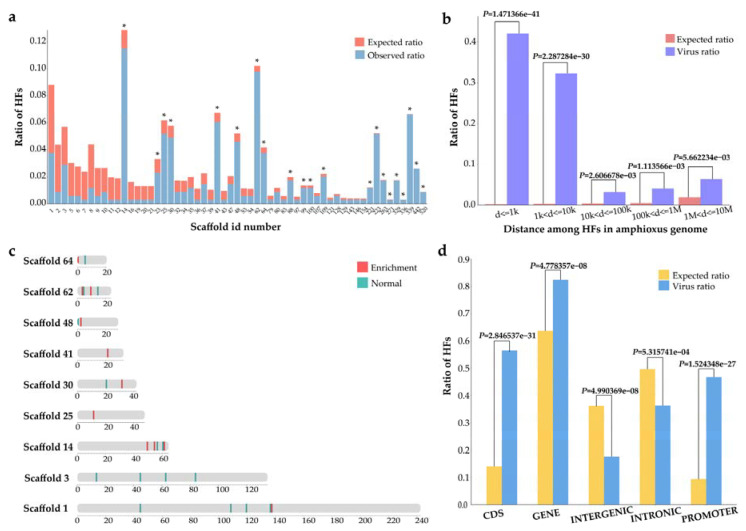
The distribution of HFs in the *B. belcheri* beihai genome. (**a**) The distribution of HFs on scaffold level. Black star represents statistically significant difference (adjusted *p* value < 0.05) between the observed number of HFs and expected (random) number of HFs in each scaffold. Adjusted *p* value were calculated by $X^{2}\;$ test (using the Bonferroni method). (**b**) The distance between neighbor HFs in the *B. belcheri* beihai genome. The number of HFs within multiple distances was calculated. A uniformly random distribution of HFs across the entire *B. belcheri* beihai genome was used to calculate the expected ratio. Pink bar shows the expected ratio of HFs. Purple bar shows the observed ration of HFs. These *p* values were calculated by $X^{2}$ test. (**c**) Distribution of HFs in each 100 kb sliding windows on scaffolds. Scaffolds with no less than 10 HFs and length greater than 1000 kb were shown in figure. Sliding windows enriched with HFs were colored as red (*p* < 0.01, $X^{2}$ test); sliding windows without enrichment of HFs were colored as green. (**d**) The distribution of HFs in functional genomic regions of the *B. belcheri* beihai genome. The expected (random distribution, yellow) and the observed (actual numbers, blue) ratios of fragments in the CDS, gene, intergenic, intron and promoter are shown. These *p* values were calculated by $X^{2}$ test.

**Figure 4 viruses-15-00909-f004:**
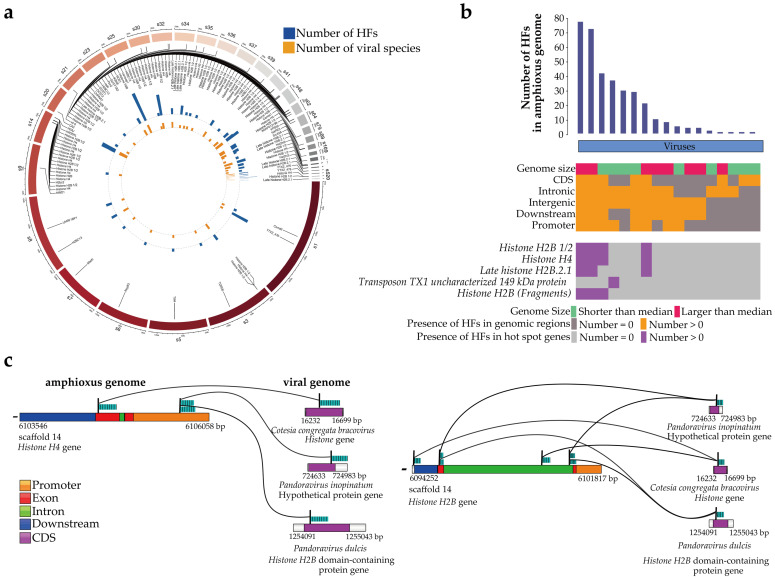
Gene and functional annotation of HFs in the *B. belcheri* beihai genome. (**a**) The number of HFs and species of virus corresponding to each gene in the *B. belcheri* beihai genome. The number in the outer circle represents the scaffold id number of the amphioxus genome, and the inner circle shows the amphioxus gene name. The height of the yellow bar represents the number of viruses species. The height of the blue bar represents the number of HFs in amphioxus. (**b**) General profile of HF in the *B. belcheri* beihai genome when mapped to viruses. Each blue vertical track represents a virus species. The height of blue vertical track represents the number of HFs in amphioxus. All panels are aligned with vertical tracks. The data are sorted by virus type, genome size, and the presence of HFs in the amphioxus CDS, intron, intergenic, downstream, and promoter. The bottom heat map shows the presence of HF in the hot spot homologous genes of the *B. belcheri* beihai genome. The hot spot genes were the 5 genes with the most HFs in the *B. belcheri* beihai genome. (**c**) The HFs of amphioxus genes containing the HFs with the highest confidence of long length and high alignment quality. The 5 HFs with the highest confidence were located in *Histone H4* and *Histone H2B* of amphioxus and were homologous to *C. congregata bracovirus*. In addition to the HFs with the top longest length and the highest alignment quality, other HFs within the gene were also shown. There were multiple viral DNA fragments were homologous to one DNA fragment of amphioxus and one viral DNA fragment were possibly homologous to multiple DNA fragments of amphioxus. The bars on the left represent the amphioxus genome; different colors represent different genomic regions of amphioxus; the dark green bars with vertical line represent the HFs of amphioxus. The bars on the right represent viral genomes; different colors represent different genomic regions of virus; the dark green bars with vertical line represent viral HFs. Yellow represents promoter regions in the amphioxus genome; red represents exon regions in the amphioxus genome; green represents intron regions in the amphioxus genome; blue represents downstream regions in the amphioxus genome; purple represents CDS regions in viral genomes.

**Table 1 viruses-15-00909-t001:** Viruses homologous to B. belcheri beihai genome.

Virus Accession	Number of HFs in Amphioxus	Virus Name
NC_021858.1	78	*Pandoravirus dulcis*
NC_026440.1	73	*Pandoravirus inopinatum*
NC_006639.1	42	*Cotesia congregata bracovirus*
NC_028045.1	37	*Tadarida brasiliensis circovirus 1*
NC_008168.1	30	*Choristoneura occidentalis granulovirus*
NC_001550.1	29	*Mason-Pfizer monkey virus*
NC_022098.1	21	*Pandoravirus salinus*
NC_026421.1	10	*Equid gammaherpesvirus 5*
NC_028094.1	8	*Chrysochromulina ericina virus*
NC_026141.2	5	*Adelie penguin polyomavirus*
NC_008603.1	4	*Paramecium bursaria Chlorella virus* FR483
NC_008724.1	4	*Acanthocystis turfacea Chlorella virus 1*
NC_019491.1	2	*Cyprinid herpesvirus 1*
NC_000852.5	1	*Paramecium bursaria Chlorella virus 1*
NC_001716.2	1	*Human betaherpesvirus 7*
NC_008094.1	1	*Y73 sarcoma virus*
NC_023006.1	1	*Pseudomonas phage PPpW-3*

**Table 2 viruses-15-00909-t002:** HFs in histone genes of amphioxus.

Virus Name	Number of HFs in Viruses	Number of HFs in Amphioxus	Virus Gene	Ratio of HFs in Viruses	Ratio of HFs in Amphioxus
*Cotesia congregate bracovirus*	9	39	*Histone*	55.4% (32.5%–59.1%)	50.4% (0.500%–88.4%)
*Pandoravirus dulcis*	5	70	*Histone H2B* domain-containing protein	25.7% (17.0%–26.4%)	49.5% (1.60%–77.7%)
*Pandoravirus salinus*	1	16	*Histone H2B* domain	23.4%	12.4% (7.80%–76.4%)
*Pandoravirus inopinatum*	3	68	hypothetical protein	--	--

Ratio of HFs in viruses was the ratio of the length of HFs in the viral histone genes. Ratio of HFs in the amphioxus was the ratio of the length of HFs in histone genes of the amphioxus. Outside brackets are the average ratios, and inside brackets are the highest and lowest ratios.

**Table 3 viruses-15-00909-t003:** The 5 HFs with highest confidence of longest length and the highest alignment quality.

Scaffold Id	The Length of HFs (bp)	Amphioxus Genes	Virus Name	Virus Genes
Scaffold 14	276	*Histone H4*	*Cotesia congregata bracovirus*	*Histone*
scaffold 14	275	*Histone H4*	*Cotesia congregata bracovirus*	*Histone*
scaffold 14	275	*Histone H2B*	*Cotesia congregata bracovirus*	*Histone*
scaffold 14	272	*Histone H4*	*Cotesia congregata bracovirus*	*Histone*
scaffold 14	271	*Histone H2B*	*Cotesia congregata bracovirus*	*Histone*

## Data Availability

The genome sequencing data, assembly, and annotation of *B. belcheri* beihai have been uploaded to NCBI database under the BioProject accession: PRJNA804338. The transcriptome data of *B. belcheri* beihai were from the BioProject accession: PRJNA310680. The viral genome sequence data used in this study were deposited in the NCBI (https://ftp.ncbi.nih.gov/genomes/Viruses/all.fna.tar.gz, accessed on 20 February 2022).

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
