# Peer review of "Genomic Analysis of Amphioxus Reveals a Wide Range of Fragments Homologous to Viral Sequences"

_viruses, 2023, doi:10.3390/v15040909_

Round 1

Reviewer 1 Report

In the present work, homologous fragments of virus were identified from the genome of B. belcheri Beihai, which were preferentially located at the CDs and promoter regions. In addition, the genes harboring a high frequency of HFs were homologous to Histone H4 or Histone H2B domain of viruses. This study looks interesting. However, some points must be improved before this manuscript can be published.

Below are my specific comments:

1.     In Materials and methods: Please provide the methods of genome sequencing and assembly.

2.     As far as I know, there is probably no interferon or receptors of interferon identified from amphioxus until now. How to explain DCST1 act as a regulator of Type I interferon signaling in amphioxus?

3.     I would strongly advise the authors to re-organize the Discussion section and make it more logical and less results description.

4.     Section 2.2: The last sentences of this paragraph probably transfer to Result section.

5.     The third paragraph of Discussion section: An Arabic numeral (less than ten) should not show at the beginning of sentence. Please change “4” to “Four”.

Author Response

Please refer to the attached Word file for the point-by-point responses, thanks.

Reviewer 2 Report

This manuscript identified some viral homologous fragments in the amphioxus genome. However, this work was not studied in sufficient depth, only the annotation and distribution of viral homologous fragments were explored. This leads to the lack of direct evidence for many conclusions in the discussion section.

Most importantly, two serious issues must be solved for this work:

1.The quality of amphioxus gene assembly was poor (583 scaffolds). In this case, how to identify whether the viral homologous fragment found in this work is the contaminating sequence?

2.Are these viral homologous fragments present in animals that have a lower evolutionary status than amphioxus? (I think it is very possible that some viral homologous fragments exist in lower species, and even this percentage may be very high.) If these viral homologous fragments are present in lower animals, then amphioxus may have simply inherited them smoothly from even lower animals. This issue must be resolved, otherwise all the views in the manuscript will be challenged.

Author Response

Please see the attachment for the point-by-point responses, thanks.

Reviewer 3 Report

In this manuscript, Du et al. sequenced, assembled, and annotated the genome of the Beihai amphioxus. Then the authors mapped multiple virus genomes to the amphioxus genome and found 347 homologous fragments (HFs). A comprehensive investigation proved that these HFs are associated with histone. I am interested in this paper and hope to see the studies following this paper.

Major:

1 The introduction section is too short compared to the whole paper. Please include an introduction to histone and relative histone research in the genome other than amphioxus.

2. The resolution of the main figures is too low. For example, in Figure 1b, and 1d, I cannot see the p-value. Please place Figure 1c horizontally rather than vertically. 

3 In section 2.4, please write down how many HFs are left in each step of filtering. 347 HFs in the amphioxus genome but only 50 HFs in 17 viral genomes. How and why the difference in number is coming from?

Minor:

1 Fig S3 is not clear to me. 3D figures did not present the ideas very well. At least the green virus in the E-value 3rd to 10th is hidden from the reader. A heatmap or 5 histograms may work. Please add caption for this figure.

Author Response

(The authors gave the same response as above.)
